🔓 | **Open Peer Review** | Mycology | Research Article

# The *Xenopyricularia zizaniicola* exhibits a genome architecture distinct to the two-speed genome

Zhenyu Fang,[1] Yuyong Li,[2] Jianqiang Huang,[1] Jianhong Wang,[1] Xiwen Lian,[2] Shuhui Lin,[2] Zonghua Wang,[1,3] Yongsheng Zhu,[4] Huakun Zheng[5]

**ABSTRACT** The fungal pathogens exhibit diverse genome architecture, which facilitates the host adaptation. Although increasing high-quality genomic data enable insights into the genome architecture of many fungal pathogens during the last decades, genomic features of many fungal species are still not fully characterized. Here, we identified a *Pyriculariaceae* family fungal strain *Xenopyricularia zizaniicola* JB-1 causing the leaf spot disease on *Zizania latifolia* and revealed its distinct genome compartment features. The fungal strain JB-1 was identified as *X. zizaniicola* based on the Koch's postulate, conidial morphology, and phylogenetic analysis. Using 2.51 Gb PacBio HiFi sequencing data, the JB-1 genome was assembled into nine contigs, five of which contain telomeric repeats at both ends. The genome size is 40,888,459 bp with an N50 of 6,431,016 bp, and a total of 9,894 protein-coding genes were predicted. BUSCO assessment demonstrated high completeness, with 754 (99.47%) of the 758 BUSCO orthologs identified as complete. The absence of both repeat-rich regions at chromosome ends and preferential residing of pathogenicity-associated genes (PAGs) in the repeat-rich regions indicated a genome compartment dissimilar to the "two-speed genome" commonly observed in *Pyricularia oryzae*, indicating a distinct evolution drive of the PAGs in *X. zizaniicola* strain JB-1. Additionally, the JB-1 genome encodes fewer PAGs compared to other members of family *Pyriculariaceae*. These findings provide valuable genomic resources of family *Pyriculariaceae* and will facilitate future studies on host-pathogen interactions and the development of effective disease management strategies for *X. zizaniicola*.

**IMPORTANCE** The family *Pyriculariaceae* includes notorious pathogens that annually result in significant agricultural losses. The genome architecture of plant fungal pathogens reflects their evolutionary adaptation to host-pathogen interactions. However, limited knowledge exists regarding the genomic features of other species within family *Pyriculariaceae*, particularly those associated with the economically important crop *Zizania latifolia*. In this study, we assembled the first high-quality genome of *Xenopyricularia zizaniicola* strain JB-1, which infects *Z. latifolia*, and revealed its distinct genome architecture. We provide evidence that the distribution pattern of pathogenicity-associated genes in *X. zizaniicola* strain JB-1 closely resembles the "one-speed genome" structure, which contrasts with *Pyricularia oryzae*. Our findings provide valuable resources for genomic studies within family *Pyriculariaceae* and contribute to our understanding of the adaptive evolution of pathogens to their hosts.

**KEYWORDS** *Xenopyricularia zizaniicola*, *Zizania latifolia*, genome assembly, pathogenicity-associated genes, one-speed genome

*Z*izania latifolia is a perennial aquatic plant originated in China and distributed in Japan, Korea, and Southeast Asia (1, 2). In China, *Z. latifolia* with swollen culms is a popular vegetable and traditional herbal medicine, commonly known as *Jiaobai* (3).

**Peer Reviewer** Alex Zaccaron, Oregon State University, Corvallis, Oregon, USA

Address correspondence to Huakun Zheng, huakunzheng@163.com, Yongsheng Zhu, zysfaas@qq.com, or Zonghua Wang, wangzh@fafu.edu.cn.

The authors declare no conflict of interest.

The cultivation of *Z. latifolia* has become an important industry in Zhejiang, Jiangsu, and other regions of China, with significant economic value (4, 5). *Xenopyricularia zizaniicola*, a fungal pathogen which belongs to family *Pyriculariaceae*, primarily infects the leaves and leaf sheaths of *Z. latifolia* (6). The disease incidence ranges from 30% to 50%, while in severe cases, it can reach as high as 70% to 80%, leading to significant yield losses in *Z. latifolia* (7).

The genome architecture of plant fungal pathogens is highly dynamic and compartmentalized, reflecting their evolutionary adaptation to host-pathogen interactions. Many filamentous plant pathogens exhibit a "two-speed genome" organization, where conserved housekeeping genes are located in gene-dense, repeat-poor regions, while genes associated with virulence, such as effectors, are enriched in gene-sparse, repeat-rich regions (8–10). These repeat-rich regions facilitate the diversification of pathogenicity-associated genes (PAGs) through mechanisms such as deletions, translocations, and repeat-induced point mutations (RIP) (11, 12). Such genomic compartmentalization has been well-documented in pathogens like *Pyricularia oryzae, Fusarium oxysporum,* and *Leptosphaeria maculans*, where virulence genes are preferentially located in repetitive, transposable elements (TEs)-rich regions (13–15). This dual-genome structure enables pathogens to balance the stability required for essential cellular functions with the flexibility needed to adapt to host defenses and environmental changes (10). Besides, the other two genome organizations were also proposed, namely one- and multiple-speed genomes. Unlike the two-speed genome architecture, one-speed genomes exhibit a more uniform distribution of genes. For example, effector genes and non-virulence-related genes of barley powdery mildew fungus are evenly spread across the genome without forming distinct compartments (9). The evolution of virulence traits in one-speed genomes is often governed by mechanisms such as copy-number variation and heterozygosity of effector loci (9).

The family *Pyriculariaceae* includes notorious pathogens like *P. oryzae*, the causative agent of rice blast, which annually results in significant agricultural losses (16). Whole-genome sequencing has emerged as a powerful tool for uncovering the genetic foundations of fungi, allowing researchers to explore their evolutionary history, host interactions, and environmental adaptations (17–19). Despite the extensive research on *P. oryzae*, there is limited knowledge regarding the genomic features of other species within family *Pyriculariaceae*, particularly those associated with aquatic plants like *Z. latifolia*.

In this study, we isolated the strain JB-1 from *Z. latifolia* and identified it as *X. zizaniicola* with a combination of morphologic, pathogenicity, and phylogenetic analyses. Then, we conducted whole-genome sequencing of *X. zizaniicola* strain JB-1, followed by *de novo* genome assembly and annotation. Finally, we performed a comparative genomic analysis between *X. zizaniicola* strain JB-1 and other members of family *Pyriculariaceae* and revealed a distinct distribution of putative PAGs in JB-1 genome. The genomic data provide valuable resources for genomic studies in family *Pyriculariaceae* and will contribute to our understanding of the adaptive evolution of pathogens to their hosts.

## RESULTS

### Identification of *X. zizaniicola* strain JB-1 causing leaf spot disease on *Z. latifolia*

We isolated fungal spores from the leaf spot lesions from *Z. latifolia* collected at Wuyishan City, Fujian Province in 2023. One of the representative strain JB-1 was used for further study. The colony of strain JB-1 was circular, white, and covered with abundant aerial hyphae (Fig. 1A). The reverse side of the colony exhibited brown pigmentation (Fig. 1B). Microscopic examination revealed that the conidiophores were long, erect, branched, and bearing conidia (Fig. 1C). The conidia were pyriform, 2-septate, and exhibited a more obovoid shape compared to the typical conidia of *Pyricularia* species (Fig. 1D). These morphological characteristics are consistent with the genus

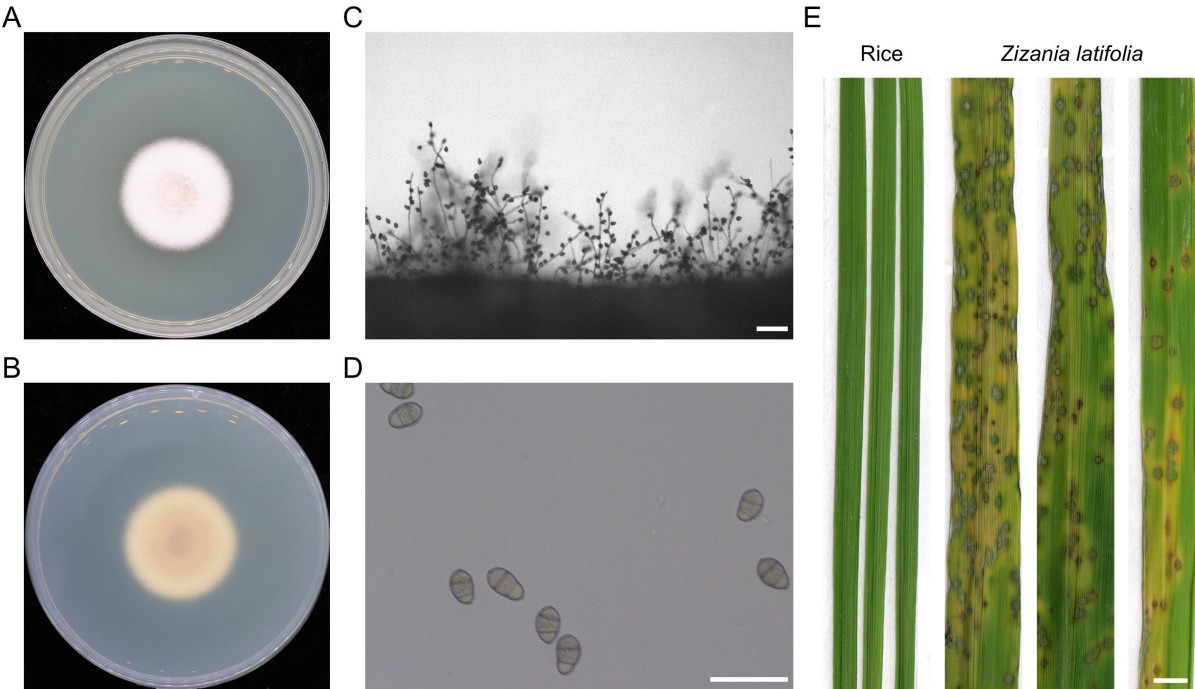

**FIG 1** Colony morphology, conidiogenesis, and pathogenicity of fungal strain isolated from leaf spot on *Z. latifolia*. (A) Front view of strain JB-1 colony grown on CM plate at 28°C for 10 days. (B) Reverse view of the same colony on CM plate. (C) Microscopic observation of conidiophores and conidia using a light microscope after 48 h of induction, scale bar = 100 μm. (D) The conidia morphology of strain JB-1, scale bar = 50 μm. (E) Pathogenicity of strain JB-1 toward rice and *Z. latifolia*. Conidial suspensions (1 × 10$^5$ conidia/mL) were sprayed onto 3-week-old *Z. latifolia* and rice seedlings. Diseased leaves were photographed at 5 days post-inoculation, scale bar = 0.5 cm.

*Xenopyricularia* (6). To assess the pathogenicity of this strain, the spore suspension was spray inoculated onto *Z. latifolia* and rice seedlings. Typical susceptible lesions were only developed from the *Z. latifolia* leaves 5 days after inoculation. The lesions were diamond-shaped or spindle-shaped, with a tan center surrounded by dark brown margins and a yellow halo (Fig. 1E). These results confirm JB-1 as a pathogenic fungus of *Z. latifolia*.

To elucidate the phylogenetic relationship of strain JB-1 with other members of family *Pyriculariaceae*, a maximum-likelihood phylogenetic tree was constructed based on concatenated sequences of ITS, Calmodulin, and RPB1. The representative genus from family *Pyriculariaceae* includes *Neopyricularia*, *Pseudopyricularia*, *Pyricularia*, *Macgarvieomyces*, *Proxipyricularia,* and *Xenopyricularia*. The strain JB-1 was clustered within the clade of *X. zizaniicola* (Fig. 2). We inferred from these results that the JB-1 strain belongs to *X. zizaniicola*.

## Genome assembly of *X. zizaniicola* strain JB-1

The genome of *X. zizaniicola* strain JB-1 was sequenced using the PacBio HiFi sequencing method and was assembled with Canu (v2.2). The final assembly comprised 9 contigs with a total genome size of 40,888,459 bp (Table 1). The assembly displayed a high level of contiguity, with an N50 of 6,431,016 bp and the max contig length of 6,886,082 bp. The GC content of the genome was 46%, and no assembly gaps were detected. Additionally, five telomere-to-telomere (T2T) contigs were identified. To evaluate genome completeness, BUSCO analysis was performed using the fungi_odb10 database. The assessment of genome completeness indicated that out of a total of 758 orthologous BUSCO genes, 754 (99.47%) were identified as complete and single-copy orthologs, no duplicated and fragmented orthologs. Only 4 (0.53%) missing orthologs have been identified, suggesting that the genome assembly is of high quality and nearly complete. To evaluate the base accuracy of the assembly, MerQury (v1.3) was used with PacBio HiFi

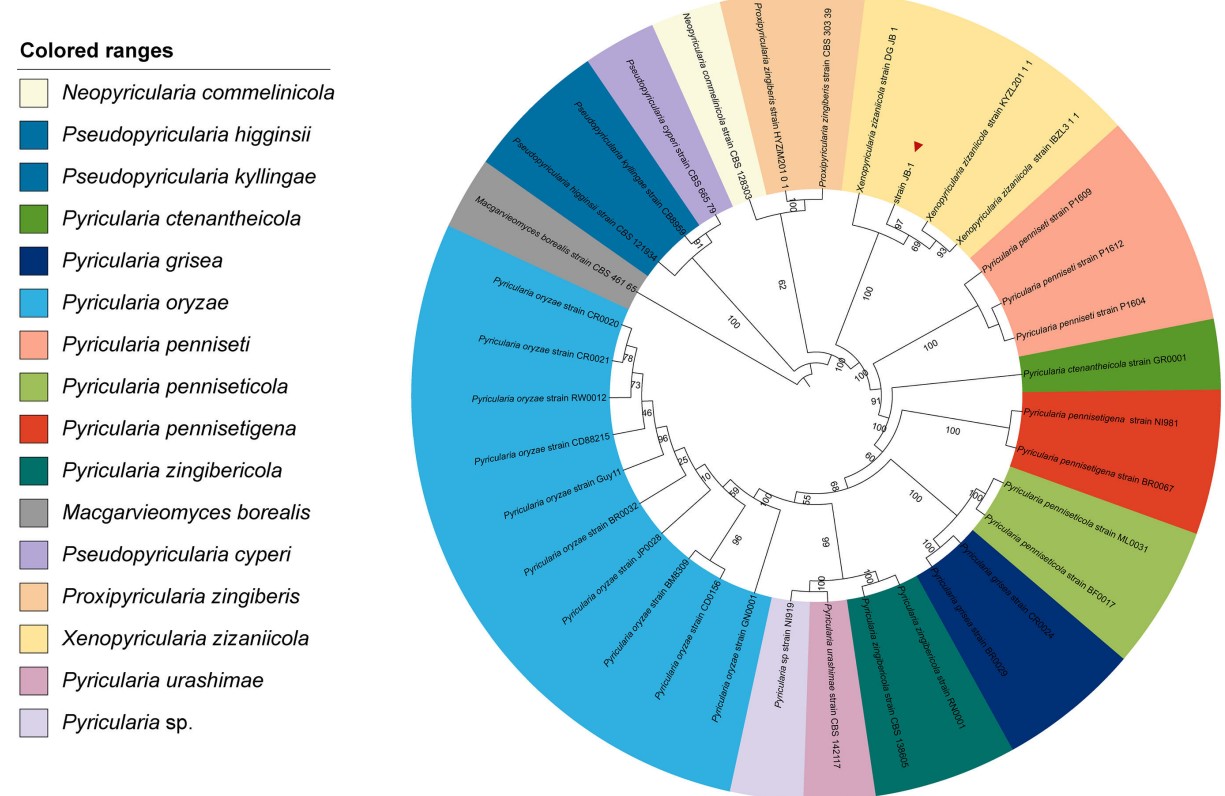

**FIG 2** Phylogenetic analysis of genus from family *Pyriculariaceae*. The phylogenetic tree constructed by maximum likelihood (ML) method based on concatenated sequences of ITS, Calmodulin, and RPB1. Each colored range represents a distinct species.

**Colored ranges**

- *Neopyricularia commelinicola*
- *Pseudopyricularia higginsii*
- *Pseudopyricularia kyllingae*
- *Pyricularia ctenantheicola*
- *Pyricularia grisea*
- *Pyricularia oryzae*
- *Pyricularia penniseti*
- *Pyricularia penniseticola*
- *Pyricularia pennisetigena*
- *Pyricularia zingibericola*
- *Macgarvieomyces borealis*
- *Pseudopyricularia cyperi*
- *Proxipyricularia zingiberis*
- *Xenopyricularia zizaniicola*
- *Pyricularia urashimae*
- *Pyricularia* sp.

reads. The k-mer analysis indicated a high level of accuracy in the JB-1 genome assembly (Table S1). We employed BRAKER2 to predict protein-coding genes, identifying a total of 9,894 genes. RepeatModeler was subsequently used to analyze repetitive sequences, which accounted for 2,105,835 bp, representing 5.15% of the genome. These repetitive sequences consisted of retroelements (92.94%), DNA transposons (5.11%), and unidentified sequences (1.95%).

## Gene functional annotation

The functional annotation of 9,894 predicted proteins was conducted using the Gene Ontology (GO) database and Kyoto Encyclopedia of Genes and Genomes (KEGG). A total of 3,705 genes were assigned GO terms, and 2,152 genes were mapped to KEGG pathways. GO analysis revealed that the top five most abundant GO terms were intracellular anatomical structure, organelle, cellular metabolic process, cytoplasm, and primary metabolic process (Fig. 3A). KEGG analysis further indicated that the top five most abundant pathways included ribosome, purine metabolism, cell cycle—yeast, nucleocytoplasmic transport, and protein processing in the endoplasmic reticulum (Fig. 3B). In addition, pathogenicity-associated genes (PAGs) were identified, including 508 carbohydrate-active enzyme (CAZyme) genes, 531 secondary metabolite biosynthesis genes (SM_genes), 606 secreted protein genes, and 331 candidate effector genes.

## The PAGs displayed a distinct distribution pattern in *X. zizaniicola* strain JB-1

To further understand the genomic features of *X. zizaniicola* strain JB-1, a synteny analysis was performed between *X. zizaniicola* strain JB-1 genome and the well-characterized *P. oryzae* strain Guy11 genome (Fig. 4). Interestingly, we observed significant differences in the distribution of repeat sequences between *X. zizaniicola* strain JB-1 and *P. oryzae* strain Guy11 (Fig. 4A). In *P. oryzae* strain Guy11, repeat sequences are preferentially

**TABLE 1** Summary of *X. zizaniicola* strains JB-1 assembly

| Statistics | *X. zizaniicola* strains JB-1 |
|---|---|
| Number contigs | 9 |
| Assembly size (bp) | 40,888,459 |
| L50 | 4 |
| Contig N50 (bp) | 6,431,016 |
| Max contig length | 6,886,082 |
| Mean contig length | 4,543,162 |
| Gap number | 0 |
| GC content (%) | 46% |
| Contigs in T2T chromosome level | 5 |
| Single-copy BUSCOs (S) | 99.47% |
| Duplicated BUSCOs (D) | 0.0% |
| Fragmented BUSCOs (F) | 0.0% |
| Missing BUSCOs (M) | 0.53% |
| Complete BUSCOs (C:S + D) | 99.47% |
| Genomic solid k-mer | 36,022,061 |
| Total solid k-mer in reads | 36,384,522 |
| k-mer completeness (%) | 99% |
| Repetitive sequence (%) | 5.15% |
| Retroelements (%) | 4.79% |
| DNA transposons (%) | 0.26% |
| Unclassified repetitive sequence (%) | 0.1% |
| Protein-coding gene number | 9,894 |

enriched in the subtelomeric regions of the chromosomes. However, this pattern is less pronounced in *X. zizaniicola* strain JB-1, especially on chromosomes 4 and 6; repeat sequences are tend to be concentrated in the internal regions rather than at the ends of the chromosomes. Moreover, there is a difference in the distribution between predicted PAGs, such as effector genes and SM_genes. In *P. oryzae* strain Guy11, these genes are predominantly located in repeat-rich regions, suggesting a potential link between repeat sequences and pathogenicity. In contrast, in *X. zizaniicola* strain JB-1, these genes are mostly found in repeat-sparse regions (Fig. 4A). To validate this observation, we calculated the intergenic size and distances between these PAGs and repeat sequences. The intergenic size revealed that both *P. oryzae* strain Guy11 and *X. zizaniicola* strain JB-1 exhibit one-compartment genome organization (Fig. S1). However, in JB-1, the distance from effector genes to repeat sequences showed no significant difference compared to those of other genes, whereas in Guy11, effector genes were significantly closer to repeat elements (Fig. S2). Moreover, compared to *P. oryzae* strain Guy11, the PAGs in *X. zizaniicola* strain JB-1 were more distant from repeat sequences (Fig. 4B and C). This distinctive genomic organization suggested a unique evolutionary pressure or adaptation in *X. zizaniicola* strain JB-1.

## Comparative genomic analysis of *X. zizaniicola* strain JB-1 and other members of family *Pyriculariaceae*

To better understand the genomic characteristics of *X. zizaniicola* strain JB-1, we obtained the genome sequences of an additional 11 species of family *Pyriculariaceae* from the NCBI genome database for comparison. The average size of the nuclear genomes of these 12 strains was 43.33 Mb, with the largest genome found in *Pyricularia pennisetigena* strain BR36 at 49.12 Mb and *X. zizaniicola* strain JB-1 had the smallest genome at 40.89 Mb (Fig. 5B). To further validate the genome size of *X. zizaniicola* strain JB-1, k-mer-based genome size estimation was performed using HiFi reads and analyzed with GenomeScope v2.0. The estimated genome size was 40.12 Mb, which is consistent with the assembled genome size. These results suggest that the genome sizes of family *Pyriculariaceae* exhibit limited variation.

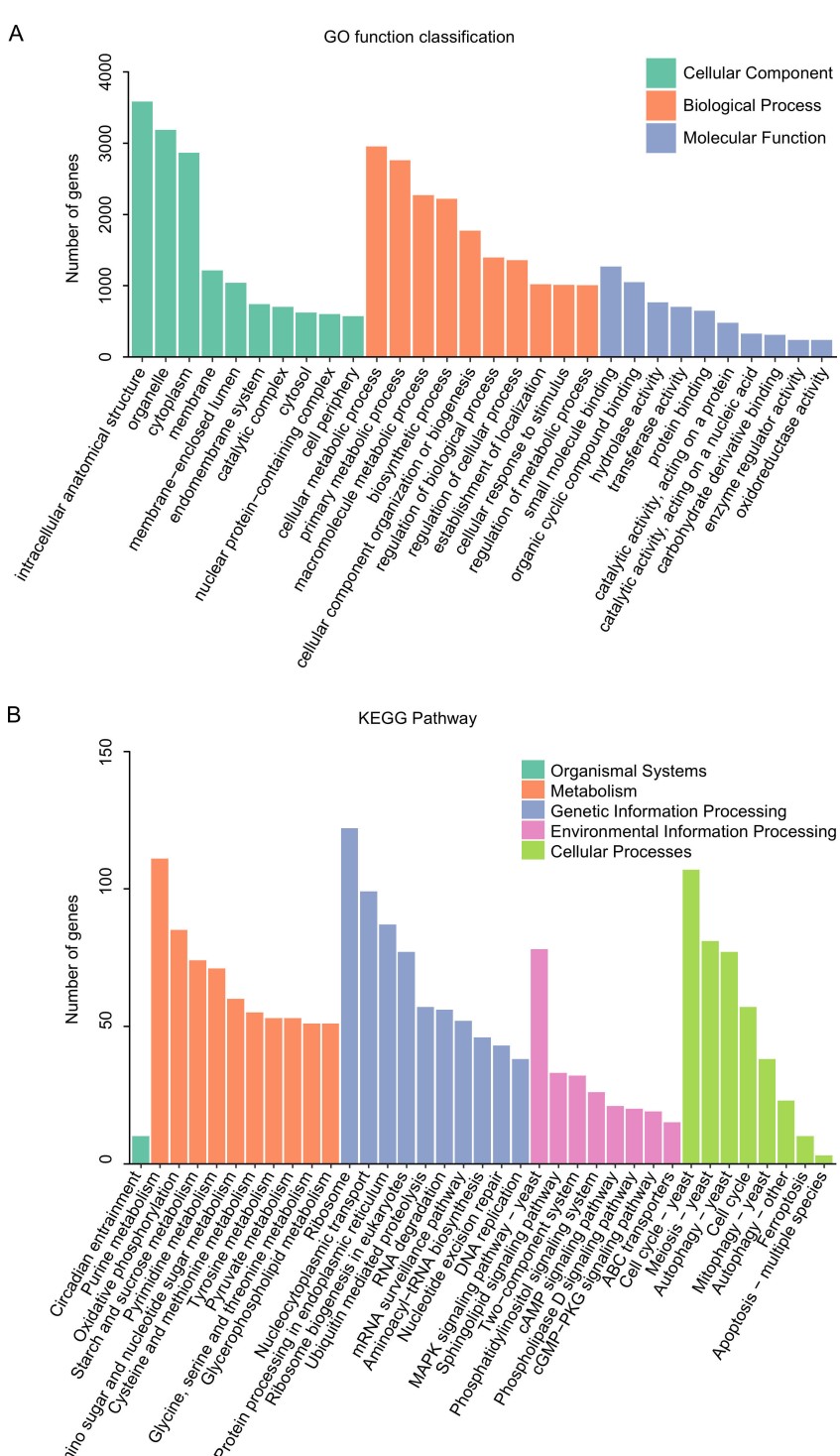

**FIG 3** Gene functional annotation of *X. zizaniicola* strain JB-1. GO annotation (A) and KEGG pathway annotation (B) of predicted proteins in *X. zizaniicola* strain JB-1.

Gene prediction was also conducted for the 12 genomes. The gene content ranged from 35.64% (*P. pennisetigena* strain PM1) to 49.16% (*P. oryzae* strain Guy11), indicating variability in gene density across the species. Notably, *X. zizaniicola* strain JB-1 has lower gene content and the smallest predicted proteome, encoding only 9,894 proteins (Fig. 5C and E).

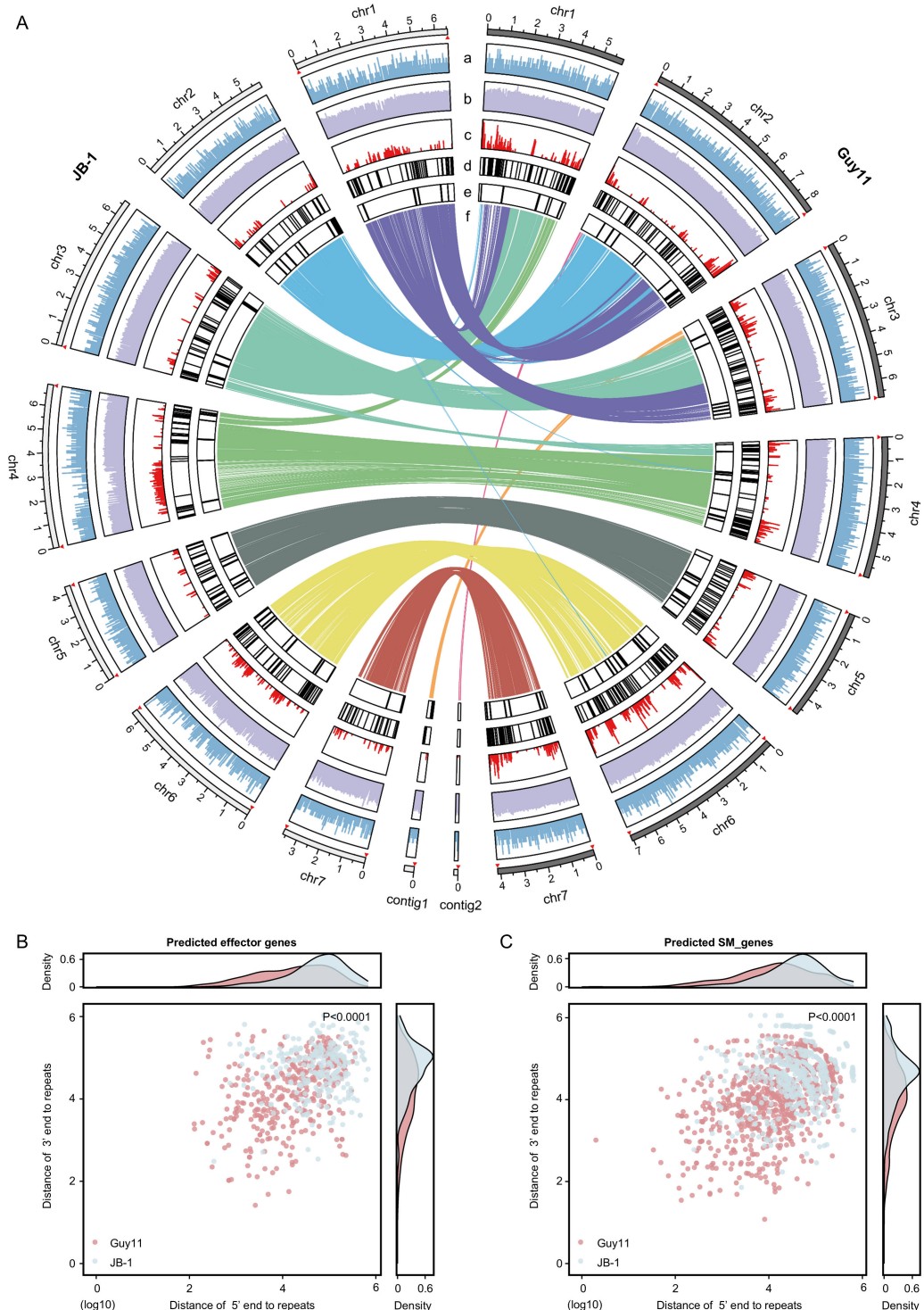

**FIG 4** Comparison of whole-genome assemblies between *X. zizaniicola* strain JB-1 and *P. oryzae* strain Guy11. (A) Comparison of genome feature. Track a shows the density of predicted genes. Track b represents GC content. Track c illustrates the density of repetitive sequences. Tracks a–c are plotted using 10 kb non-overlapping windows, with the *y*-axis scaled from zero to the maximum. Tracks d and e display the distribution of effector genes and secondary metabolite biosynthesis genes (SM_genes), respectively. Track f shows the syntenic regions. Red triangles mark the locations of telomeres. (B) The 5′ and 3′ ends distance of predicted effector genes to repeats. (C) The 5′ and 3′ ends distance of predicted SM_genes to repeats. Statistical analysis was performed using the Mann–Whitney *U* test.

Repetitive sequences were identified using a consistent pipeline across the 12 strains, revealing an average repetitive sequence content of 7.14%. Among 12 strains, *P. oryzae* Guy11 had the highest proportion of repetitive sequences (12.53%), while *Magnaporthales* sp. strain P1609 had the lowest (4.49%) (Fig. 5D).

Further analysis identified potential PAGs in each strain, including carbohydrate-active enzymes (CAZymes), secondary metabolite biosynthesis gene clusters (SM_clusters), secreted proteins, and effectors (Fig. 5F through I). On average, the 12 *Pyriculariaceae* species contained 600 CAZymes, 63 SM_clusters, 796 secreted proteins, and 449 effectors. *X. zizaniicola* strain JB-1 exhibited a lower number of these PAGs compared to other species in family *Pyriculariaceae*, which reflects distinct evolutionary adaptations and pathogenic strategies.

## DISCUSSION

Although numerous species from family *Pyriculariaceae* have been sequenced and their genome data published, the existing genomic resources remain insufficient to fully capture the genetic diversity within the family *Pyriculariaceae*. In this study, strain JB-1 was isolated and identified as *X. zizaniicola* based on cultural characteristics, conidial morphology, phylogenetic analysis, and pathogenicity assays (Fig. 1 and 2). It was confirmed as the causal pathogen of *Z. latifolia*. The genome of *X. zizaniicola* strain JB-1 was assembled using PacBio HiFi sequencing data. The assembly consists of nine contigs, five of which contain telomeric repeats at both ends, indicating high assembly quality (Table 1). These genome sequences contribute to the enrichment of *Pyriculariaceae* genomic resources.

CAZymes, secondary metabolites, and effector proteins produced by phytopathogenic fungi play crucial roles in host-pathogen interactions (20–22). Studies on the genomes of phytopathogenic fungi have demonstrated that PAGs tend to reside in genomic regions enriched with transposable elements that exhibit accelerated rates of sequence evolution (23). These rapidly evolving genomic regions are described by the term "two-speed genome" (8). In *P. oryzae* strain Guy11, repetitive sequences are predominantly enriched at the chromosomal ends, where putative PAGs preferentially reside. In contrast, neither the enrichment of repetitive sequences at the subtelomeric region nor the preferential residing of putative PAGs in repeat-rich regions was observed in the genome of *X. zizaniicola* strain JB-1 (Fig. 4). The distribution pattern of PAGs in *X. zizaniicola* strain JB-1 closely resembles the "one-speed genome" structure. Similar distribution patterns had been observed in nonclassically secreted effectors lacking signal peptides (24, 25). These results indicated that distinct mechanisms have evolved for the variation of different types of effector genes or effector genes in different fungal species. In addition, the patterns of "one-speed genome" have been observed in obligate

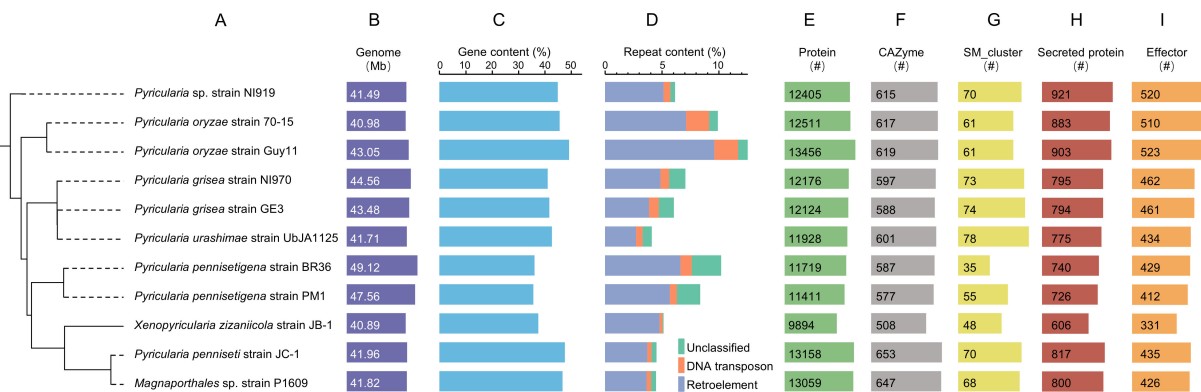

**FIG 5** Phylogenomic relationships and genomic features of 8 species from family *Pyriculariaceae*. (A) A maximum likelihood phylogenomic tree of 11 assemblies from family *Pyriculariaceae*. (B) Genome sizes. (C) Proportions of gene content. (D) Percentage of three categories of repeats. (E–I) Number of predicted proteins (E), CAZymes (F), SM_clusters (G), secreted proteins (H), and effectors (I).

biotrophic fungi such as rust fungi and powdery mildew fungi (26–30). In obligate biotrophic fungi, certain effectors are crucial for survival because successful colonization of the host is essential for completing their life cycle. As a result, selection pressure in obligate biotrophs might promote recombination events that link key effectors with other essential genes, ensuring their inheritance and conservation (28). This divergence suggests that *X. zizaniicola* strain JB-1 appears to rely on a more stable genomic organization. This may limit its ability to rapidly adapt to new host defenses but could reflect a finely tuned interaction with its specific host.

Among the 12 species of family *Pyriculariaceae*, *X. zizaniicola* strain JB-1 possesses fewer potential PAGs. This reflects the influence of host-specific resistance, driving the evolution of PAGs in strains adapted to different hosts. Rather than relying on a wide array of virulence factors, *X. zizaniicola* strain JB-1 has specialized in more efficient mechanisms for interacting with its host. The lower number of PAGs could reflect its high degree of adaptation to particular host, possibly indicating a narrow host range.

## Conclusion

In this study, we assembled a high-quality genome of the *X. zizaniicola* strain JB-1 infecting the economically important crop *Z. latifolia* in China and revealed its distinct genome architecture. Our findings provide valuable insights into the genomic features and evolutionary adaptations of *X. zizaniicola* and will facilitate future investigations into host-pathogen interactions and the development of effective management strategies for *X. zizaniicola*.

## MATERIALS AND METHODS

### Isolation of *X. zizaniicola* strain JB-1

Leaf spot lesions from *Z. latifolia* were collected in Wuyishan City, Fujian Province (27.610445°N, 118.21317°E) in 2023. To isolate fungal strains, lesions were surface-sterilized (75% ethanol, 60 s), rinsed with sterile water, and placed on moist filter paper in a petri dish. The dish was incubated under light conditions at 28°C for 72 h to induce conidiation. After incubation, spores were gently scraped from the lesions using a sterile toothpick and streaked onto water agar plates. Single spores were then isolated under the microscope and transferred to complete medium (CM: 6 g/L yeast extract, 6 g/L acid-hydrolyzed casein, 10 g/L sucrose, 20 g/L agarose) to obtain purified strains.

### Growth, conidiation, and pathogenicity analysis of *X. zizaniicola* strain JB-1

To observe the colony morphology, the strain was cultured on CM at 28°C in the dark. After 10 days, the colony was photographed using a Nikon camera. For conidiation analysis, strains were grown on rice bran medium (RBM: 40 g/L rice bran, pH 6.0–6.5) at 28°C in the light for 10 days. Conidia were harvested and suspended in distilled water; the morphology of the conidia was examined and photographed using an Olympus BX53 microscope. For pathogenicity analysis, conidial suspensions ($1 \times 10^5$ conidia/mL in 0.02% Tween 20) from *X. zizaniicola* strain JB-1 strains were sprayed on 3-week-old *Z. latifolia* and rice seedlings. Disease symptoms were observed and recorded 5 days after inoculation.

### Phylogenetic analysis

By consulting literature related to the genus *Pyricularia* (6), we selected published and highly credible reference sequences for use in the phylogenetic analysis of this study. *Macgarvieomyces borealis* CBS 461.65 was chosen as the outgroup species. DNA sequence alignment was performed using Clustalx v.1.83 (31), followed by sequence correction with BioEdit. Conserved sequences were then extracted using the Glock option in PhyloSuite v.1.2.1 (32). Subsequently, the three single-gene sequences were

combined for analysis by clicking on the Concatenate Sequence option. The best-fit nucleotide substitution models and related parameters for each gene were calculated using the AIC (Akaike Information Criterion) standard in Partitionfinder2. Afterward, IQ-TREE (33) was used to construct a maximum likelihood tree for the combined three-gene data set, and MrBayes (34) was employed to build a Bayesian phylogenetic tree for the same combined data set. Finally, the three single-gene sequences were concatenated using the same method to construct a phylogenetic tree for the combined three-gene data set.

To construct a phylogenetic tree based on orthologous gene clusters, high-quality genome sequences and annotation files of 10 *Pyricularia* species were retrieved from the NCBI database with the following filtering criteria: N50 >1 Mb, sequencing depth ≥50×, and BUSCO completeness >90%. Genome completeness was evaluated using BUSCO v5.5.0 (35). Transposable elements (TEs) were annotated and filtered from all predicted genes using TEsorter v2.0 (36) (parameters: -eval 1e-5 -score 0.6), followed by the removal of TE-associated genes. The longest transcript of each gene, along with corresponding protein sequences, CDS sequences, and updated annotation files, was extracted from the filtered gene set using TBtool-II v2.042 (37) (parameters: -minCDS 300 -maxStop 0). OrthoFinder v2.6.6 (38) (parameters: -M msa -S diamond -T fasttree -t 16a 10) was employed to identify orthologous gene clusters, from which single-copy orthologs shared across all 10 species were selected for subsequent analyses. For each gene cluster, multiple sequence alignment was performed using MAFFT v7.505 (39) (parameters: --localpair --maxiterate 1000), followed by trimming of low-quality regions with trimAl v1.5.0 (40) (parameter: -automated1). The maximum likelihood phylogenetic tree was constructed using IQ-TREE (33) based on the optimal substitution model determined by ModelFinder (41) (parameters: -m TESTNEW -mfreq FU -mtree), with node support assessed through 1,000 ultra-fast bootstrap replicates (parameters: -B 1000 -bnni). The resulting phylogeny was visualized using FigTree v1.4.4 (http://tree.bio.ed.ac.uk/software/figtree/).

## Whole-genome sequencing

Genomic DNA of *X. zizaniicola* strain JB-1 was extracted with the DNeasy Plant Mini Kit (QIAGEN), and its integrity was assessed using the Agilent 4200 Bioanalyzer (Agilent Technologies, Palo Alto, California). DNA was sheared using g-Tubes (Covaris) and purified with AMPure PB magnetic beads. SMRTbell libraries were constructed with the Pacific Biosciences SMRTbell Template Prep Kit 1.0, size-selected for 20 kb molecules using Sage ELF, and sequenced on the PacBio Sequel II platform for 30 h at Annoroad Gene Technology Co., Ltd. PacBio HiFi reads were processed using Fastp v0.23.1 to remove adapter contamination, low-quality bases (Phred quality <5), and unrecognizable nucleotides (42).

## Genome assembly and evaluation

PacBio reads were assembled using Canu v2.2 (43) with the "-pacbio-hifi" option, only PacBio reads longer than 5 kb were used for the assembly. Assembly completeness was assessed using BUSCO v5.5.0 with the fungi_odb10 database (35). Assembly quality was assessed using Merqury v1.3 (44). Based on 17-mer frequency distribution generated by Jellyfish v2.3.0 (45), genome size was estimated using GenomeScope v2.0 (46). Telomeres were detected with the TIDK v.0.2.0 (47), employing the telomeric sequence (TTAGGG/CCCTAA) (48) and the following parameters: tidk explore—minmum 5—maximum 12 genome.fa tidk search—string TTTAGGG --dir outdir—output output genome.

## Gene and repetitive sequence annotation

Protein-coding genes were annotated using the BRAKER 2.0 pipeline (49), integrating ab initio gene predictions from AUGUSTUS v3.4.0 (50) and GeneMark-EP+ (51), as well as homology evidence from the OrthoDB fungal database. Repetitive sequences were

annotated using both *ab initio* and homology-based methods. An *ab initio* transposable element library was constructed with RepeatModeler (52), and RepeatMasker v4.1.5 (53) was used to perform a homology-based repeat search throughout the genome.

## Gene functional annotation

All predicted proteins were annotated using eggNOG-Mapper (54), followed by classification of Gene Ontology (GO) terms and Kyoto Encyclopedia of Genes and Genomes (KEGG) pathways using the dplyr package in R. The results were visualized by ggplot2. dbCAN2 was employed for the prediction of carbohydrate-active enzymes (CAZymes) (55). Secondary metabolite biosynthetic gene clusters (SM_clusters) were predicted using antiSMASH (56). Signal peptide and transmembrane domain prediction was performed using SignalP v6.0 and TMHMM 2.0, respectively (57, 58). Proteins shorter than 400 amino acids, containing signal peptides, and predicted to lack transmembrane domains were classified as secreted proteins. Effector proteins were further identified using EffectorP v3.0 (59). The number of CAZymes, SM_clusters, secreted proteins, and effectors was visualized using the ChiPlot (https://www.chiplot.online/).

## Comparative genomic analysis

Collinearity analysis was performed using One Step MCScanX in TBtools-II v2.042 (47). The distribution of genes, GC content, repetitive sequences, effector genes, secondary metabolite biosynthetic genes (SM_genes), and collinearity between *X. zizaniicola* strain JB-1 and *P. oryzae* strain Guy11 were illustrated using the Advanced Circos tool in TBtools-II v2.042 (47).

## ACKNOWLEDGMENTS

This research was supported by the National Key Research and Development Program of China (2023YFD1400201) and the Science and Technology Major Program of Fujian Province, China (2022NZ030014).

## AUTHOR AFFILIATIONS

[1]State Key Laboratory of Agricultural and Forestry Biosecurity, College of Plant Protection, Fujian Agriculture and Forestry University, Fuzhou, Fujian, China
[2]College of Life Sciences, Fujian Agriculture and Forestry University, Fuzhou, Fujian, China
[3]Fuzhou Institute of Oceanography, Minjiang University, Fuzhou, Fujian, China
[4]Rice Research Institute, Fujian Academy of Agricultural Sciences, Fuzhou, Fujian, China
[5]National Engineering Research Center of JUNCAO Technology, College of Juncao Science and Ecology, Fujian Agriculture and Forestry University, Fuzhou, Fujian, China

## AUTHOR ORCIDs

Zhenyu Fang  http://orcid.org/0009-0002-2524-0388
Zonghua Wang  http://orcid.org/0000-0002-0869-9683
Yongsheng Zhu  http://orcid.org/0000-0003-4723-0017
Huakun Zheng  http://orcid.org/0000-0001-5274-2717

## DATA AVAILABILITY

The raw genomic sequencing data used during the current study are available at NCBI Sequence Read Archive database (Accession numbe SRR32895135) (60). The assembled genome was deposited under the same BioProject with *Xenopyricularia zizaniicola* strain JB-1 at NCBI (Accession number: JBNCJZ000000000, BioProject ID: PRJNA1242696) (61, 62).

## ADDITIONAL FILES

The following material is available online.

### Supplemental Material

**Figures S1 and S2 (Spectrum00362-25-s0001.pdf).** Fig. S1: The intergenic length. Fig. S2: Distance from genes to repeats.
**Supplemental table (Spectrum00362-25-s0002.xlsx).** Base accuracy of JB-1 assembly.

### Open Peer Review

**PEER REVIEW HISTORY (review-history.pdf).** An accounting of the reviewer comments and feedback.

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
