## [Reviewer comments · Microbiology Spectrum]

Microbiology Spectrum

The *Xenopyricularia zizaniicola* exhibits a genome architecture distinct to the two-speed genome

Zhenyu Fang, Yuyong Li, Jianqiang Huang, Jianhong Wang, Xiwen Lian, Shuhui Lin, Zonghua Wang, Yongsheng Zhu, and Huakun Zheng

Corresponding Author(s): Huakun Zheng, Fujian Agriculture and Forestry University

Review Timeline:

Submission Date:	February 5, 2025
Editorial Decision:	March 22, 2025
Revision Received:	April 17, 2025
Accepted:	May 3, 2025

Editor: Matthew Anderson

Reviewer(s): Disclosure of reviewer identity is with reference to reviewer comments included in decision letter(s). The following individuals involved in review of your submission have agreed to reveal their identity: Alex Zaccaron (Reviewer #2)

Transaction Report:

DOI: <https://doi.org/10.1128/spectrum.00362-25>

Re: Spectrum00362-25 (The *Xenopyricularia zizaniicola* exhibits a novel genome architecture distinct to the two-speed genome)

Dear Dr. Huakun Zheng:

Thank you for the privilege of reviewing your work. Below you will find my comments, instructions from the Spectrum editorial office, and the reviewer comments.

Two major areas of concern have been brought up that are important for the implications of this work. The first is the comparison to genome with similar numbers of repeat to demonstrate the two-speed hypothesis. The second major concern to address includes aspects of the genome description, namely the duplication of many genes in the BUSCO analysis. Please make sure to also deposit the genome sequence into a publicly accessible database.

Revision Guidelines

Sincerely,
Matthew Anderson
Editor
Microbiology Spectrum

Reviewer #1 (Comments for the Author):

This manuscript present a HiFi based near-complete genome assembly of the fungal strain *Xenopyricularia zizaniicola* JB-1

which causing leaf spot on *Zizania latifolia*, a Chinese wild rice and well-known as the second-largest aquatic vegetable 'Jiaobai' in China.

Comments

1. Abstract, suggest to add more detail of genome assembly like genome size, N50, gene number, and genome quality like BUSCO completeness for reader got key genome feature of this fungi.
2. Similar, suggested added more genome feature in Table 1.
3. How about the average base accuracy of genome assembly JB-1? Suggest add a Merqury accuracy valuation with HiFi reads.
4. *X. zizaniicola* JB-1 have the smallest genome size shown in Fig 5A. Suggest to add a k-mer based genome size estimation using HiFi reads to check it.
5. Lines 61-64, It's not clear what is the host *Zizania latifolia*, suggest add more introduction about *Z. latifolia*, actually it's a Chinese wild rice used as cereal crops and traditional medicine food, and well-known as vegetable 'Jiaobai'.
6. Lines 270-272, The precise geographic coordinates where the isolate collected should be included.
7. Fig. 1E, the leaf of *Z. latifolia* seems much bigger than rice, them were merged from different photos? Suggest to add a bar and make sure them in original size.
8. Fig. 2, what the number on the branch node means?
9. Fig. 4, what dose SM_genes means, suggest to added a note. It seems effectors and SM genes have more distance to repeats but not clear enough, suggest add a statistic P-value to enhance the conclusion. Please check X and Y subtitle, is it 'end to repeats'?
10. Fig. 5, What is the difference of the phylogenomic tree in Fig 5A and Fig 2? It seems using whole genome gene set, if so, please add more detail information including the accession and database like GenBank of the genome assemblies and also the pipeline and software used in method section. Please check the title, there is only 11 assemblies and 8 species used, not 12.

Reviewer #2 (Comments for the Author):

In this study, authors present the genome of *Xenopyricularia zizaniicola*, a fungal pathogen of *Zizania latifolia*. This is a new representative genome of a pathogen from the Pyriculariaceae family, that includes important pathogens of grasses. Authors confirmed the pathogenicity toward *Z. latifolia* and the phylogenetic placement of the strain sequenced. Comparative analyses with close relatives indicated that *Z. zizaniicola* has a small gene content and overall low abundance of repetitive DNA. Moreover, authors report that *X. zizaniicola* has a genome architecture that better resembles the "one-speed genome" model of evolution that contrasts with the close relative *Pyricularia oryzae*. The results are interesting, and the genome is an important resource to the *Xenopyricularia* genus within Pyriculariaceae family. However, I have a major and some minor concerns and comments as described below.

My major concern is about one of the main conclusions of the study, that is *X. zizaniicola* has a "one-speed genome" model of evolution. Authors support this conclusion by comparing the distance of pathogenicity-associated genes (PAG) to nearest repeats between *X. zizaniicola* and *P. oryzae*. However, this comparison is not fit for genomes that vary considerably in repeat content. For example, *P. oryzae* has twice as much repetitive DNA than *X. zizaniicola*. To better support this conclusion, I think the authors should add a comparison of intergenic size of genes, which is more standard to evaluate one-speed and two-speed genomes (see <https://doi.org/10.1111/mpp.12738> and <https://doi.org/10.1093/gbe/evaa267>).

Some of my minor concerns are about text polishing. I think authors should overall improve the text in many parts of the manuscript.

1. I think the title needs improvement: "a novel genome architecture distinct to the two-speed genome". The architecture is not novel; the one-speed genome organization has been described in other fungal species.
2. Line 22: "The fungal pathogens evolve diverse genome compartment to facilitate the host adaptation". This sentence is not accurate. It is true that the genome architecture can facilitate adaptation to the host, but there is lack of literature supporting that the genome architecture evolve for this purpose.
3. Line 25: "genomic features of many fungal species remain uncovered". I think what the authors mean is not "uncovered".
4. Line 64: The sentence "a fungal pathogen belongs to family Pyriculariaceae" is missing a relative pronoun.
5. Line 73-74: In "These repeat-rich regions facilitating the diversification of pathogenicity-associated genes (PAGs)", The word "facilitating" does not fit here. perhaps replace with "facilitate".
6. Line 79: First time using abbreviation "TE", spell out "transposable elements" here.
7. Line 82: "the other two genome compartments were also proposed, namely one- and multiple-speed genomes". The literature commonly refers to "compartments" as regions of the genome that are TE-rich or TE-poor. What authors are referring here are other genome organization or compartmentalization.

8. Line 109: I suggesting changing "caused" by "causing" in the title
9. Line 130: The genus "Pseudopyricularia" was mentioned twice.
10. Line 145: The overall BUSCO completeness is good, but 10.5% duplication rate is unusually high when running BUSCO in genome mode. Have the authors noted any long-duplicated segment in the genome, or long regions that do not align to *P. oryzae* Guy11?
11. Line 148: In the results, authors say that "We employed GeneMark to predict protein-coding genes". I suggest mentioning BRAKER2 instead of GeneMark to be more precise.
12. Lines 158-163: Authors refer to enriched GO terms and KEGG pathways. However, no comparison was made to support enrichment. What authors are reporting here appears to be simply GO terms and KEGG categories with highest number of assigned genes. Referring to as "enrichment" is confusing. Also, authors should add more details here, e.g., how many genes had GO terms assigned? How many were assigned to KEGG pathways? I also feel that authors should report here number of genes in other functional categories, such as candidate effectors, SM genes, and CAZymes.
13. Line 181-183: "The results revealed that, compared to *P. oryzae* strain Guy11, the PAGs in *X. zizaniicola* strain JB-1 were more distant from repeat sequences.". This could be due to difference in TE content between *X. zizaniicola* and *P. oryzae*. I suggest including a comparison of intergenic size of genes to demonstrate that, in *X. zizaniicola*, the intergenic size of PAG genes does not deviate from the rest of the genome.
14. Line 271-274: It's difficult to understand the methods for strain collection. For example, this sentence is incomplete "Place the lesions in a humid environment under light conditions for 3 days". I suggest the authors to improve the language and details of this paragraph.
15. Lines 299-301: Authors should add references for IQ-TREE and MrBayes
16. Line 300: This sentence is confusing: "to construct a maximum likelihood tree for the combined two-gene dataset". Weren't three genes used for the phylogenetic analysis?
17. Line 322: The "ith the fungi_odb10 database" needs revision.
18. Line 334: The tool name "RepeatModele" needs revision
19. Lines 338-339: Details are missing here for GO and KEGG pathway annotation. Which tools were used to assign GO and KEGG annotation?
20. Fig 5: I suggest including the genome-level BUSCO result in this figure for all genomes. I think it would be more informative than, for example, % gene content. Particularly if *X. zizaniicola* indeed has an unusual high number of duplicated BUSCOs.
21. The genome and the HiFi reads of *X. zizaniicola* should be made publicly available.

Dear Editor,

We would like to sincerely thank you for your time and effort in reviewing our manuscript "The *Xenopyricularia zizaniicola* exhibits a genome architecture distinct to the two-speed genome". We greatly appreciate your insightful comments, which have significantly helped us improve the quality of our paper. We have addressed all the points raised by the reviewers and have made revisions accordingly. Below are our point-by-point responses to each of the reviewers' comments.

Reviewer #1:

1. Abstract, suggest to add more detail of genome assembly like genome size, N50, gene number, and genome quality like BUSCO completeness for reader got key genome feature of this fungi.

Response: Thank you for the valuable suggestion. We agree that including key genome assembly metrics in the Abstract will better highlight the quality and features of this fungal genome. In the revised manuscript, we have added the following details to the Abstract: "The genome size is 40,888,459 bp with an N50 of 6,431,016 bp, and a total of 9,894 protein-coding genes were predicted. BUSCO assessment demonstrated high completeness, with 754 (99.47%) of the 758 BUSCO orthologs identified as complete."

2. Similar, suggested added more genome feature in Table 1.

Response: Thank you for the valuable suggestion. We have now expanded Table 1 to include additional genome features. The added features are: L70, Contig N70, Genomic solid k-mer, total solid k-mer in reads and k-mer completeness.

3. How about the average base accuracy of genome assembly JB-1? Suggest add a Merqury accuracy valuation with HiFi reads.

Response: Thank you for the valuable suggestion. We have performed Merqury analysis using HiFi reads to evaluate the base accuracy of the JB-1 genome assembly. The quality metrics have been added to Supplementary Table 1 as follows:

Chromosome	Unique k-mers (assembly-only)	Shared k-mers (assembly + reads)	QV Score	Error Rate
chr1	6	6,544,009	72.6814	5.39E-08
chr2	26	5,831,577	65.8126	2.62E-07
chr3	1	6,689,814	80.5586	8.79E-09
chr4	19	6,886,066	67.8967	1.62E-07
chr5	7	4,304,373	70.1926	9.57E-08
chr6	24	6,431,000	66.5852	2.20E-07
chr7	13	3,622,590	66.7552	2.11E-07
contig1	8	412,052	59.4231	1.14E-06
contig2	0	166,834	+inf	0

4. *X. zizaniicola* JB-1 have the smallest genome size shown in Fig 5A. Suggest to add a k-mer based genome size estimation using HiFi reads to check it.

Response: Thank you for the helpful suggestion. As recommended, we performed a k-mer based genome size estimation by GenomeScope version 2.0. The following descriptions have been added: "To further validate the genome assembly, k-mer-based genome size estimation was performed using HiFi reads and analyzed with GenomeScope v2.0. The estimated genome size was 40.12 Mb, consistent with the assembled genome size."

5. Lines 61-64, It's not clear what is the host *Zizania latifolia*, suggest add more introduction about *Z latifolia*, actually it's a Chinese wild rice used as cereal crops and traditional medicine food, and well-known as vegetable 'Jiaobai'.

Response: Thank you for the valuable suggestion. We have added descriptions about *Z latifolia*: In China, *Z. latifolia* with swollen culms is a popular vegetable and traditional herbal medicine, commonly known as *Jiaobai*.

6. Lines 270-272, The precise geographic coordinates where the isolate collected should be included.

Response: Thank you for pointing this out. We have added the geographic coordinates in the manuscript: Leaf spot lesions from *Z. latifolia* were collected in Wuyishan City, Fujian Province (27.610445°N, 118.21317°E) in 2023.

7. Fig. 1E, the leaf of *Z. latifolia* seems much bigger than rice, them were merged from different photos? Suggest to add a bar and make sure them in original size.

Response: Thank you for the valuable suggestion. The leaves of *Z. latifolia* and rice are both in the same photo. The leaves of *Z. latifolia* are indeed larger than rice, and bar has been added.

8. Fig. 2, what the number on the branch node means?

Response: The numbers on the branch node represent the bootstrap support values. This value indicates the reliability of the branch, values $\geq 70\%$ are considered to be reliable.

9. Fig. 4, what dose SM_genes means, suggest to added a note. It seems effectors and SM genes have more distance to repeats but not clear enough, suggest add a statistic P-value to enhance the conclusion. Please check X and Y subtitle, is it 'end to repeats'?

Response: Thank you for the valuable suggestion. We have implemented the following revisions in Fig. 4:

(1) Note for SM_gende has been added: secondary metabolite biosynthesis genes

(SM_genes).

- (2) P value has been added and Statistical analysis was performed using the Mann - Whitney U test.
- (3) 'prime to repeats' in X and Y subtitle has been modified to 'end to repeats'.

10. Fig. 5, What is the difference of the phylogenomic tree in Fig 5A and Fig 2? It seems using whole genome gene set, if so, please add more detail information including the accession and database like GenBank of the genome assemblies and also the pipeline and software used in method section. Please check the title, there is only 11 assemblies and 8 species used, not 12.

Response: Thank you for pointing this out. The phylogenomic tree in Fig 5A was constructed based on orthologous gene clusters. We have implemented the following revisions in Fig. 5:

(1) As suggested, we added more details to the Methods section:

To construct a phylogenetic tree based on orthologous gene clusters, high-quality genome sequences and annotation files of ten *Pyricularia* species were retrieved from the NCBI database with the following filtering criteria: N50 > 1 Mb, sequencing depth $\geq 50\times$, and BUSCO completeness >90%. Genome completeness was evaluated using BUSCO v5.5.0. Transposable elements (TEs) were annotated and filtered from all predicted genes using TEsorter v2.0 (parameters: -eval $1e-5$ -score 0.6), followed by the removal of TE-associated genes. The longest transcript of each gene, along with corresponding protein sequences, CDS sequences, and updated annotation files, were extracted from the filtered gene set using TBtool-II v2.042 (parameters: -minCDS 300 -maxStop 0).

OrthoFinder v2.6.6 (parameters: -M msa -S diamond -T fasttree -t 16 -a 10) was employed to identify orthologous gene clusters, from which single-copy orthologs shared across all ten species were selected for subsequent analyses. For each gene cluster, multiple sequence alignment was performed using MAFFT v7.505 (parameters: --localpair --maxiterate 1000), followed by trimming of low-quality regions with trimAl v1.5.0 (parameter: -automated1). The maximum likelihood phylogenetic tree was constructed using IQ-TREE based on the optimal substitution model determined by ModelFinder (parameters: -m TESTNEW -mfreq FU -mtree), with node support assessed through 1,000 ultra-fast bootstrap replicates (parameters: -B 1000 -bnni). The resulting phylogeny was visualized using FigTree v1.4.4 (<http://tree.bio.ed.ac.uk/software/figtree/>).

(2) The title was revised to "FIG 5 Phylogenomic relationships and genomic features of 8 species from family *Pyriculariaceae*. (A) A maximum likelihood phylogenomic tree of 11 assemblies from family *Pyriculariaceae*."

Reviewer #2:

1. Authors support this conclusion by comparing the distance of pathogenicity-associated genes (PAG) to nearest repeats between *X. zizaniicola* and *P. oryzae*. However, this comparison is not fit for genomes that vary considerably in repeat content. For example, *P. oryzae* has twice as much repetitive DNA than *X. zizaniicola*. To better support this conclusion, I think the authors should add a comparison of intergenic size of genes, which is more standard to evaluate one-speed and two-speed genomes (see <https://doi.org/10.1111/mpp.12738> and <https://doi.org/10.1093/gbe/evaa267>).

Response: Thank you for the valuable suggestion. As suggested, we calculated intergenic distances for *X. zizaniicola* JB-1 and *P. oryzae* Guy11. The genomes of *X. zizaniicola* JB-1 and *P. oryzae* Guy11 both exhibit one-compartment genome organization (Fig.S1). However, in *X. zizaniicola* JB-1, the distance from effector genes to repeat sequences showed no significant difference compared to those of other genes, whereas in Guy11, effector genes were significantly closer to repeat sequences (Fig.S2). This result support the hypothesis that *X. zizaniicola* JB-1 follows a one-speed genome model, while *P. oryzae* Guy11 aligns with the two-speed genome model.

2. I think the title needs improvement: "a novel genome architecture distinct to the two-speed genome". The architecture is not novel; the one-speed genome organization has been described in other fungal species.

Response: Thank you for the suggestion. The title was revised to "The *Xenopyricularia zizaniicola* exhibits a genome architecture distinct to the two-speed genome".

3. Line 22: "The fungal pathogens evolve diverse genome compartment to facilitate the host adaptation". This sentence is not accurate. It is true that the genome architecture can facilitate adaptation to the host, but there is lack of literature supporting that the genome architecture evolve for this purpose.

Response: Thank you for pointing this out. The original statement has been revised to: "The fungal pathogens exhibit diverse genome architecture, which facilitate the host adaptation."

4. Line 25: "genomic features of many fungal species remain uncovered". I think what the authors mean is not "uncovered".

Response: Thank you for pointing this out. The original statement has been revised to: "genomic features of many fungal species are still not fully characterized."

5. Line 64: The sentence "a fungal pathogen belongs to family Pyriculariaceae" is missing a relative pronoun.

Response: Thank you for the kind reminding. The original statement has been revised to: "a fungal pathogen which belongs to family *Pyriculariaceae*".

6. Line 73-74: In "These repeat-rich regions facilitating the diversification of pathogenicity-associated genes (PAGs)", The word "facilitating" does not fit here. perhaps replace with "facilitate".

Response: We sincerely appreciate this grammatical correction. The "facilitating" has been revised to "facilitate".

7. Line 79: First time using abbreviation "TE", spell out "transposable elements" here.

Response: Thank you for pointing this out. The "TE" has been corrected into "transposable elements (TEs)"

8. Line 82: "the other two genome compartments were also proposed, namely one- and multiple-speed genomes". The literature commonly refers to "compartments" as regions of the genome that are TE-rich or TE-poor. What authors are referring here are other genome organization or compartmentalization.

Response: Thank you for pointing this out. The original statement has been revised to: "Besides, the other two genome organizations were also proposed, namely one- and multiple-speed genomes."

9. Line 109: I suggesting changing "caused" by "causing" in the title

Response: Thank you for the suggestion. The " caused " has been corrected into "causing" in the title.

10. Line 130: The genus "Pseudopyricularia" was mentioned twice.

Response: Thank you for pointing this out. We have removed the redundant mention of "Pseudopyricularia".

11. Line 145: The overall BUSCO completeness is good, but 10.5% duplication rate is unusually high when running BUSCO in genome mode. Have the authors noted any long-duplicated segment in the genome, or long regions that do not align to *P. oryzae* Guy11?

Response: We sincerely apologize for the mistake. The previously reported 10.5% duplication rate was due to the use of an incorrect version of the genome for the BUSCO analysis. We have re-analyzed the data using the correct genome version and have updated the results:

Single-copy BUSCOs(S)	99.47%
Duplicated BUSCOs(D)	0.0%
Fragmented BUSCOs(F)	0.0%
Missing BUSCOs(M)	0.53%
Complete BUSCOs(C:S+D)	99.47%

12. Line 148: In the results, authors say that "We employed GeneMark to predict protein-coding genes". I suggest mentioning BRAKER2 instead of GeneMark to be more precise.

Response: Thank you for your suggestion. We have revised the original statement by replacing "GeneMark" with "BRAKER2".

13. Lines 158-163: Authors refer to enriched GO terms and KEGG pathways. However, no comparison was made to support enrichment. What authors are reporting here appears to be simply GO terms and KEGG categories with highest number of assigned genes. Referring to as "enrichment" is confusing. Also, authors should add more details here, e.g., how many genes had GO terms assigned? How many were assigned to KEGG pathways? I also feel that authors should report here number of genes in other functional categories, such as candidate effectors, SM genes, and CAZymes.

Response: Thank you for your suggestion. Following your recommendation, we have revised the relevant section and added more details:

The functional annotation of 9,894 predicted proteins was conducted using the Gene Ontology (GO) database and Kyoto Encyclopedia of Genes and Genomes (KEGG). A total of 3,705 genes were assigned GO terms, and 2,152 genes were mapped to KEGG pathways. GO analysis revealed that the top five most abundant GO terms were intracellular anatomical structure, organelle, cellular metabolic process, cytoplasm, and primary metabolic process (Fig. 3A). KEGG analysis further indicated that the top five most abundant pathways included ribosome, purine metabolism, cell cycle – yeast, nucleocytoplasmic transport and protein processing in the endoplasmic reticulum (Fig. 3B). In addition, pathogenicity-associated genes (PAGs) were identified, including 508 carbohydrate-active enzyme (CAZyme) genes, 531 secondary metabolite biosynthesis genes (SM_genes), 606 secreted protein genes, and 331 candidate effector genes.

14. Line 181-183: "The results revealed that, compared to *P. oryzae* strain Guy11, the PAGs in *X. zizaniicola* strain JB-1 were more distant from repeat sequences.". This could be due to difference in TE content between *X. zizaniicola* and *P. oryzae*. I suggest including a comparison of intergenic size of genes to demonstrate that, in *X. zizaniicola*, the intergenic size of PAG genes does not deviate from the rest of the genome.

Response: Thank you for your suggestion. As suggested, we analyzed the intergenic distances of pathogenicity-associated genes (PAGs) in *X. zizaniicola* JB-1 and *P. oryzae* Guy11. The results showed that, in both species, the intergenic distances of PAGs do not significantly differ from those of other genes in the genome (Fig. S1). Additionally, we examined the distances between genes and repeat sequences. In *X. zizaniicola* JB-1, there was no significant difference between effector genes and other genes, while in *P. oryzae* Guy11, effector genes were significantly closer to repeat sequences (Fig. S2). This result indicated that effector genes in *X. zizaniicola* JB-1 do not deviate from the rest of the genome.

15. Line 271-274: It's difficult to understand the methods for strain collection. For example, this sentence is incomplete "Place the lesions in a humid environment under light conditions for 3 days". I suggest the authors to improve the language and details of this paragraph.

Response: Thank you for pointing this out. We have revised the paragraph by providing more details of the fungal isolation procedures:

To isolate fungal strains, lesions were surface-sterilized (75% ethanol, 60 s), rinsed with sterile water, and placed on moist filter paper in a petri dish. The dish was incubated under light conditions at 28°C for 72 hours to induce conidiation. After incubation, spores were gently scraped from the lesions using a sterile toothpick and streaked onto water agar plates. Single spores were then isolated under microscope and transferred to complete medium (CM: 6 g/L yeast extract, 6 g/L acid-hydrolyzed casein, 10 g/L sucrose, 20 g/L agarose) to obtain purified strains.

16. Lines 299-301: Authors should add references for IQ-TREE and MrBayes

Response: Thank you for your suggestion. We have added appropriate references for IQ-TREE and MrBayes in the revised manuscript.

17. Line 300: This sentence is confusing: "to construct a maximum likelihood tree for the combined two-gene dataset". Weren't three genes used for the phylogenetic analysis?

Response: Thank you for pointing this out. You are correct — three genes were used in the phylogenetic analysis. We have revised the sentence.

18. Line 322: The "ith the fungi_odb10 database" needs revision.

Response: Thank you for pointing this out. We have corrected the typo error in the sentence.

19. Line 334: The tool name "RepeatModele" needs revision

Response: Thank you for pointing this out. The "RepeatModele" has been corrected into "RepeatModeler".

20. Lines 338-339: Details are missing here for GO and KEGG pathway annotation. Which tools were used to assign GO and KEGG annotation?

Response: Thank you for your suggestion. We have added details about the tools used for GO and KEGG annotation:

All predicted proteins were annotated using eggNOG-Mapper (v2.1.12), followed by classification of Gene Ontology (GO) terms and Kyoto Encyclopedia of Genes and Genomes (KEGG) pathways using the dplyr package in R. The results were visualized using ggplot2.

21. Fig 5: I suggest including the genome-level BUSCO result in this figure for all genomes. I think it would be more informative than, for example, % gene content. Particularly if *X. zizaniicola* indeed has an unusual high number of duplicated BUSCOs.

Response: Thank you for your suggestion. The high number of duplicated BUSCOs in our original analysis was due to the use of an incorrect genome version. After re-performing the BUSCO analysis with the correct genome version, the issue was resolved, and the results no longer show an unusually high number of duplicated BUSCOs. Therefore, the genome-level BUSCO results was removed from Figure 5.

22. The genome and the HiFi reads of *X. zizaniicola* should be made publicly available.

Response: Thank you for pointing this out. The raw genomic sequencing data of *X. zizaniicola* have been deposited in NCBI Sequence Read Archive database (Accession number SRR32895135). The assembled genome was deposited under the same BioProject with *Xenopyricularia zizaniicola* strain JB-1 at NCBI (Accession number: JBNCJZ000000000, BioProject ID: PRJNA1242696).

Re: Spectrum00362-25R1 (The *Xenopyricularia zizaniicola* exhibits a genome architecture distinct to the two-speed genome)

Dear Dr. Huakun Zheng:

Your manuscript has been accepted, and I am forwarding it to the ASM production staff for publication. Your paper will first be checked to make sure all elements meet the technical requirements. ASM staff will contact you if anything needs to be revised before copyediting and production can begin. Otherwise, you will be notified when your proofs are ready to be viewed.

In the profs, please consider the following changes for Table 1:

1. Contig N70 and L70 are unusual, suggest use N90 and L90 to replace it or just delete them.
2. T2T number suggested replace with 'Contigs in T2T chromosome level'

Sincerely,
Matthew Anderson
Editor
Microbiology Spectrum

Reviewer #1 (Comments for the Author):

All of my comments have been well addressed.

Here have small comments for the revision in Table 1:

1. Contig N70 and L70 are unusual, suggest use N90 and L90 to replace it or just delete them.
2. T2T number suggested replace with 'Contigs in T2T chromosome level'